# The KERES Ontology: Protecting Cultural Heritage from Extreme Climate Events

**Jürgen Reuter *** 📷, **Tobias Hellmund** 📷, **Jürgen Moßgraber** 📷 and **Philipp Hertweck** 📷

Fraunhofer IOSB, 76131 Karlsruhe, Germany; tobias.hellmund@iosb.fraunhofer.de (T.H.);
juergen.mossgraber@iosb.fraunhofer.de (J.M.); philipp.hertweck@iosb.fraunhofer.de (P.H.)
*** Correspondence: juergen.reuter@iosb.fraunhofer.de

**Abstract:** Protecting and preserving cultural heritage *(CH)* in view of global climate change is the main objective of the KERES project. For managing climate impact with proper measurements, including prevention and responsive actions, an ontology has been devised in the course of this project in close cooperation with relevant stakeholders of selected CH assets that serve us as case studies. In particular, the ontology supports modeling specific CH assets with respect to the challenges of climate change. It turns out the main challenge is to subsume the diversity of models and processes specific to individual assets on a proper level of abstraction. Based on the ontology, we succeeded in creating software that assists stakeholders in managing their CH challenges, including an interactive app for suggesting preventative measurements and a web application for creating *route cards* that are used by emergency service professionals in the case of rescuing cultural assets. We are confident that our methodology of CH assets abstraction and modeling will be applicable to a broader range of CH assets.

**Keywords:** cultural heritage *(CH)*; climate change; ontology; OWL; KERES

## 1. Introduction

Recently, cultural heritage *(CH)* faces several new types of threats from global crises, including direct and indirect threats from climate change, war, and pandemic incidents. The KERES project (*Kulturgüter vor Extremklimaereignissen schützen und Resilienz erhöhen*, Engl. *protecting cultural assets from extreme climate events and increasing resilience*) [1] combines new detailed climate forecasts and interdisciplinary analysis of criticality with proven procedures of prevention, adaptation, and resilience options to develop emergency measures and crisis management, also in cooperation with rescue services. We draw much inspiration for the KERES project from previous work in the HERACLES project [2], which has the main objective of designing, validating, and promoting responsive systems and solutions for effective resilience of CH against climate change effects.

As part of the KERES project, we collect, store, and make available all relevant information on threats, cultural assets, and all involved stakeholders to all involved parties for effectively addressing the new threats with preventive, responsive, and recovering actions. The key issue here is that these parties are extremely heterogeneous, ranging such as from cultural asset managers, facility managers, over gardeners, and firefighters, to financiers, politicians, and many more, that need to intercommunicate and share and exchange relevant information.

This is where the ontological description comes into play for modeling all relevant aspects of endangered cultural assets and measurements for protection to support intercommunication between all participating stakeholders and to provide an application programming interface *(API)* for creating, updating, and retrieving all information. The actual challenge of the KERES ontology is to bring together all of the stakeholders and agree upon a common model based on a best-effort strategy, and implement an ontology

that utilizes this model. For this purpose, the KERES project brings together stakeholders of case studies ranging from large buildings, such as the Cologne Cathedral, over small buildings, such as the historic rural buildings in the Bad Windsheim open-air museum, to parks, such as the Sanssouci Park.

Building upon the KERES ontology, the actual benefit of the project for stakeholders and for the protection of CH manifests in various user applications. A customized web server acts as an information management system for browsing all information stored in the ontology as well as hosting further content. Furthermore, we are developing solutions tailored to the stakeholders' individual needs. For example, one of these solutions suggests preventive, responsive, and recovering actions based on input about a cultural asset's state from the user. Another example is an application for creating and managing route cards that can be printed out for use by emergency service professionals for cultural asset salvation in the case of an emergency. Some of these applications have been integrated into the information management system, as appropriate.

While developing solutions for the specific assets of the KERES case studies, we always had in mind general applicability, such that we are confident that the solutions will be easily adaptable for a broad range of other cultural assets.

## 2. Paper Outline

Considering state of the art in the affected fields of the domain (Section 3.1), the KERES project (Section 3.2) differs from similar projects mainly in that it additionally focuses on *protection* of CH, given new threats from climate change. It deploys a methodology (Section 3.3) based on the project's case studies (Section 3.4) as well as on workshops and surveys with all of the respective stakeholders in the project (Section 3.5) combined with extensive experience from the preceding HERACLES project (Section 3.6) to explore the state of the art and gather upcoming requirements in CH protection. The first results were informally recorded as a mind map (Section 3.7).

The main focus of the present work is the creation of the KERES ontology (Section 3.8), based on the mind map as well as on the heavily revised core of the HERACLES ontology (Section 3.8.2), and extending it with support for a number of new modularized features beyond HERACLES, such as a generic bookmarks (Section 3.8.3) feature, ROIs, and POIs (Section 3.8.4), tree cadasters (Section 3.8.6), and a catalog of best practice examples (Section 3.8.7).

To prove the usefulness of our work, we implemented a set of handy tools that successfully deploy the KERES ontology (Section 4), including presentation and providing all of the gathered information backed by the WEBGENESIS information management system (Section 4.1), a new application for suggesting measurements for CH (Section 4.2), and another new application that supports CH institutions in creating and managing route cards (Section 4.3).

We argue that our tools can be useful for protecting CH and discuss open issues to be addressed in the future (Section 5).

## 3. Materials and Methods

The KERES project is built around a set of CH assets that serve as case studies. An initial series of workshops and stakeholder surveys aimed at first collecting relevant terminology and concepts to be covered by the ontology. The results of these meetings were recorded in a collaboratively created mind map. Moreover, we could exploit and build upon the ontology of the preceding HERACLES project, whose main objective was to *"design, validate and promote responsive systems/solutions for effective resilience of CH against climate change effects"* [3].

### 3.1. Related Work

The KERES ontology integrates three major areas of interest (Figure 1):

- CH

- emergency & crisis management
- climate change

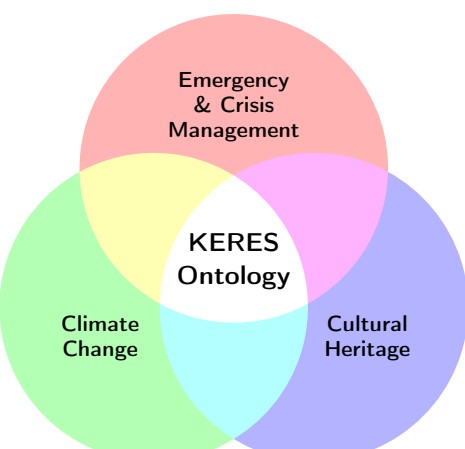

**Figure 1.** Purpose & Ontological Scope of the KERES Project.

For each of these areas, various approaches for ontological modeling already exist. The actual challenge is to combine these areas such that synergy effects can be drawn. We build on previous work where it appears feasible.

### 3.1.1. Cultural Heritage

Ontologies on CH already have an eventful history. An early and maybe still the most prevalent standard is the CIDOC *Conceptual Reference Model* [4–6], also known as ISO 21127:2014 standard. While this ontology introduces a rich set of basic concepts like `Place`, `Actor`, `Event`, `Time-Span` and useful properties for interlinking, we need enhanced support for data structures from an engineer's perspective. For example, our concepts of regions of interest (ROIs) and points of interest (POIs) do not only feature flat collections of places but also support the definition of hierarchies of regions (such as a building containing floors and the floors containing rooms) and *paths* of POIs even across regions (Section 3.8.4). Moreover, our focus is on *protection* of cultural assets and crisis management rather than documenting the history of culture. Consequently, our ontology contains concepts such as, for example, `Threat`, `Damage`, and `Preventive Measurement`, that are not covered by the CIDOC Conceptual Reference Model. Still, CIDOC's temporal concepts may be used for documenting the chronology of a cultural asset's state before and after a damage or response or recovery action, though this usage does not seem to be the original intent of CIDOC's temporal concepts. While CIDOC emerged before the advent of OWL, an OWL implementation of CIDOC has meanwhile been developed [7].

RANJGAR et al. [8] present an ontology specifically for POIs. The KERES ontology supports POIs as well. However, while they focus on quantitative modeling of POIs based on geographical coordinates with the GEOSPARQL standard [9,10], our spatial entities are often lacking geographical coordinates but are built rather upon a qualitative containedness relation. Consequently, we differentiate between POIs and ROIs, with the former used for specifying paths across regions, while the latter span tree-like structures corresponding to the containedness relation (Section 3.8.4). For the future, we may consider integrating the KERES ontology with the region connection calculus RCC8 [11], which is supported in GEOSPARQL.

### 3.1.2. Emergency & Crisis Management

While in the domain of CH, the CIDOC effort has created a widely accepted standard, we do not know of any comparable effort in the area of emergency & crisis management. In contrast, there exists a large number of individual efforts for ontologies in this area

that focus on peculiar needs that are addressed by their respective authors, although with conceptual overlaps.

More specifically, Liu, Brewster, and Shaw present a thorough survey of 26 existing ontologies for emergency and crisis management as of the year 2013 [12]. They claim that 65% of the existing ontologies are semantically interoperable (though without going much into the subtle details) and identify a set of key areas that comprehend

- process-related concepts (crisis response and disaster management)
- people
- organizations
- types of damage
- disaster
- critical infrastructure
- geography
- hydrology
- meteorology
- topographical concepts

In fact, all of these areas are also addressed by the KERES ontology, though with a focus on the fields of domain most relevant for and tailored on the applications of the KERES project for gaining the best results out of the ontology. Moreover, we could build upon prior experience with ontological modeling from our BEAWARE project that we established specifically in the domain of crisis management [13,14].

A more recent ontological effort in the area of emergency and crisis management is the *EDXL-RESCUER* ontology [15], focusing on short-term communication and messaging in acute emergency cases, whereas the KERES project focuses on medium-term processes of protection and resilience.

The *DoRES Three-tier Ontology for Modelling Crises* [16] focuses on document-centric communication around the generic key classes *documents*, *reports*, *situations* and *events*, however, obviously without further formally specializing in specific topics and issues. In contrast, the KERES project builds on an elaborated hierarchy of ontological concepts that cover a wide range of cases with a plethora of specific classes for formal modeling.

*EmergencyFire* [17] provides much more specific concepts with a design methodology much more similar to our approach. However, EmergencyFire focuses on fire events. In contrast, our scope covers a wider range of extreme climate events and also other areas such as measurement, analysis, prevention, and response.

The *Hazard and Emergency Response Ontology* (*HERO*) [18] focuses on the description of past, current, or potential future hazards. It is tailored for interoperability between applications with a special focus on the areas *human observations from direct response exercises*, *geographical information*, *data flow*, *logistical information*, *sensor data*, and *social data*, centered around the key concepts *event*, *disaster*, and *hazard*. Obviously, only the concept *hazard* is further diversified by more specific subclasses, with extreme climate events only playing a small supporting role, if at all. Once again, while there are certain overlaps with the needs of the KERES project, there are also severe differences, for example, KERES addressing cultural assets, climate, and diversifying extreme climate events.

The KERES ontology can be considered as the successor of the HERACLES ontology [2]. Though, we still had to revamp major parts (Section 3.8.2) to get them smoothly integrated with new aspects and challenges from the KERES project.

The bottom line, there are many more ontologies with overlapping vocabulary in the area of emergency and crisis management. While we acknowledge and value all of these approaches and try to draw inspiration from these efforts, none of them actually matches the wide range of areas that the KERES project addresses *and* at the same time, the level of diversity of vocabulary that we need to formally express the circumstances, processes, and needs of the KERES project.

### 3.1.3. Climate Change

While research on climate change has been an ongoing effort for decades, we could not identify any existing ontology that has been or looks promising for being accepted as an ontological standard. In fact, climate research involves many disciplines ranging, for example, from meteorology to geophysics, chemistry, biology, ecology, and many more areas, such that it seems unlikely that a single well-structured, comprehensive ontology that covers all significant aspects will ever be established.

In the fashion of a survey, ESBJÖRN-HARGENS [19] collects terminology and examples in the research area of climate change. While this work does not result in a formal description such as an OWL ontology, it informally contributes a list of 18 example research roles, associated actions, and objectives, that appear well-suited for incorporation as sample data into a formal ontology.

PILEGGI and LAMIA present an OWL ontology for documenting the course of events in climate change [20], while we focus on assessing to what extent cultural assets are endangered by extreme weather and climate events by detecting security risks and suggest adaptation strategies to be developed.

There are also efforts to automatically create ontologies from data mining by scanning large amounts of texts and extracting prominent keywords and relations to create concepts and properties [21]. While such work may obviously gain new insights into the overall topic, in the KERES project, we manually create our ontology based on engineering principles.

### 3.2. The KERES Project

The KERES project aims at developing emergency measures and crisis management, also in cooperation with rescue services. It combines new detailed climate forecasts and interdisciplinary analysis of criticality with proven procedures of prevention, adaptation, and resilience options. Climate fact sheets for a selected set of CH sites (Section 3.4) are created and, together with information of any kind emerging from the project, are collected and provided with a semantic knowledge base server featuring a formal description of all of this knowledge, based on an ontology written with the web ontology language OWL (Section 3.8). Accordingly, ontology plays a central role at the core of the whole project. The knowledge base server includes various apps that assist in knowledge networking and procedures for early warnings, such as an app for deriving recommended actions for preventive measurements or another app that assists in creating route cards. The KERES project also considers running through use cases for emergency and risk prevention. A national heritage expert panel has been established during the project to bring together associated partners and international experts, including professionals from research, state administration staff, and practitioners from emergency services. As project partners, the KERES project includes the Climate Service Center Germany (GERICS), the Fraunhofer Institute of Optronics, System Technologies and Image Exploitation IOSB (*Fraunhofer-Institut für Optronik, Systemtechnik und Bildauswertung*, Fraunhofer IOSB) [22] and the Prussian Palaces and Gardens Foundation Berlin-Brandenburg (*Stiftung Preußische Schlösser und Gärten Berlin-Brandenburg*, SPSG) [23] as consortium members, as well as the Federal Agency for Technical Relief (*Bundesanstalt Technisches Hilfswerk*, THW) [24] as external partner.

### 3.3. Methodology

Given the domain and scope of the KERES ontology directly by the project goals and reusing parts of the HERACLES ontology, we created a mind map from interviews with domain experts. Next, we created a class hierarchy and added properties. Finally, we populated the ontology with instances related to the KERES case studies, thereby also proving and evaluating the usefulness of the ontological model (Figure 2).

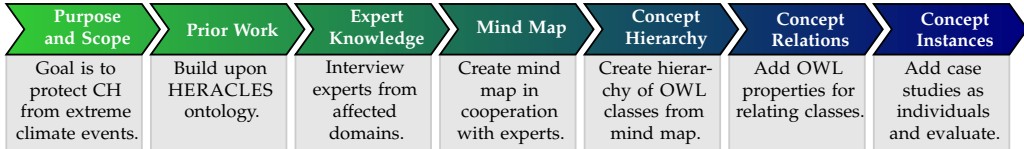

**Figure 2.** Methodology for Designing and Implementing the KERES Ontology.

1.  Purpose and Scope
    The KERES project goal of protecting CH from extreme climate events sets the domain and scope of the ontology. Specifically, it covers the description of

    - cultural assets
    - material that cultural asset consists of
    - stakeholders
    - extreme weather and climate events as a potential threat
    - potential and actual damage
    - technology for measuring and analyzing damage
    - prevention measurement
    - responsive action
    - technology for applying responsive action

    using the entities from five case studies as sample assets.

2.  Reuse of Prior Work
    The KERES ontology builds upon the HERACLES ontology from an earlier project that focuses on risk management and improving resilience against climate events. While the HERACLES ontology served well for taking over basic concepts, it turned out that many details needed refinement as well as completely new concepts and properties to match additional requirements of the KERES project. We also considered incorporating third-party ontologies such as CIDOC but found it inadequate for our purposes.

3.  Gathering Knowledge by Interviewing Experts
    Part of the KERES project is close collaboration with experts from the project partners that cover various domains. We interviewed stakeholders related to these case studies (Section 3.4), as well as additional external stakeholders (e.g., the external project partner THW).

4.  Mind Map
    Based on the interviews and in cooperation with the project partners, we created an informal mind map (Section 3.7).

5.  Concept Hierarchy
    From the mind map, we derived a class hierarchy for the ontology and integrated it with the class hierarchy of the HERACLES ontology (Section 3.8.2).

6.  Concept Relations
    Furthermore, we considered the results of the HERACLES project, and, in particular, the HERACLES ontology, which already provided essential object and datatype properties. From the mind map and HERACLES ontology, a new KERES ontology was created.

7.  Concept Instances
    Finally, we populated the ontology with instances that prototypically describe selected aspects of the KERES case studies. We validated, improved, and evaluated the ontology by implementing small proof-of-concept applications.

*3.4. KERES Case Studies*

The KERES project focuses on five individual CH sites that serve as case studies for developing concepts for CH protection (Figure 3):

- Speicherstadt (*warehouse district)* in Hamburg;
- Cologne Cathedral in Cologne

- Sanssouci Park with Charlottenhof Castle, and Babelsberg Park in Potsdam
- Franconian open-air museum in Bad Windsheim
- the small chapel Frauenbergkapelle in Sufferloh

## KERES Case Studies

The KERES case studies form the data basis of the project. Scattered accross Germany, they provide information on the effects and consequences climate change may have on historic gardens, churches, castles, palaces, and other cultural assets. Navigate either directly via the links on the map or the table below to the individual studies.

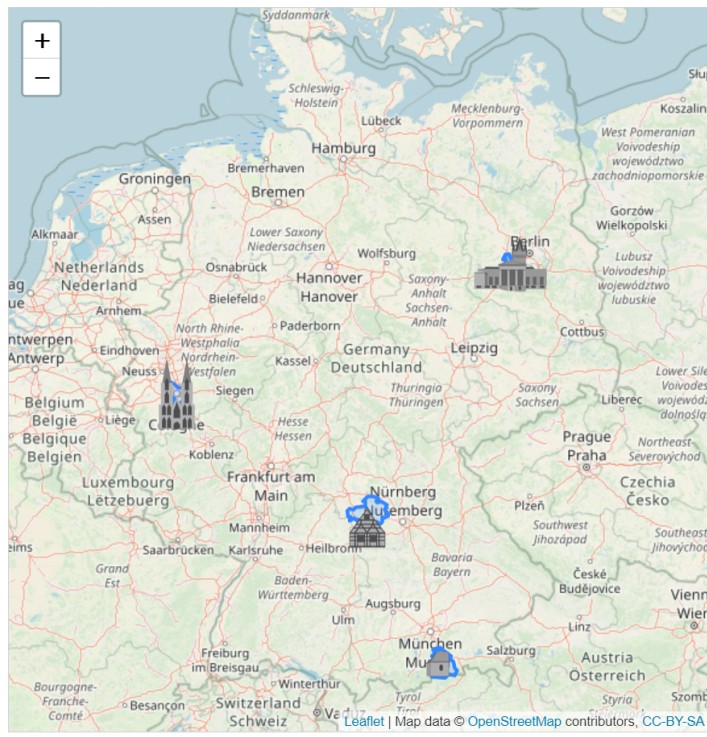

**Figure 3.** KERES case studies include cultural heritage in Hamburg, Cologne, Potsdam, Bad Windsheim, and Sufferloh (screenshot of web browser view of the project website). Map data © OpenStreetMap contributors, CC-BY-SA.

From these case studies, as two examples, we choose the Sanssouci Park and the Franconian Open-Air Museum to take a closer look at them briefly.

### 3.4.1. *Park Sanssouci* with *Castle Charlottenhof* in Potsdam, Brandenburg

Climate change is considered the main reason for an increasing number of extreme climate events in the past few years. Such events pose an existential threat, particularly for the parks of project partner SPSG. SPSG is a public foundation formed by the Prussian State Palaces and Gardens Administration. It is responsible for running and preserving CH such as parks and castles located in the Berlin-Brandenburg area. Prolonged drought, heavy rainfall, and gale-force storms cause massive damage to woody plants and infrastructure, resulting in a substantial threat to visitors of the historical cultural assets.

To avoid acute damage situations and prevent safety threats, the KERES project elaborates on exemplary measurements against threats, using the examples of Park Sanssouci and Park Babelsberg, focusing on increasing woody plants' vitality and sustainably securing the paths through the parks.

For this purpose, sensors record data on soil moisture in the exterior of the parks and feed it into the knowledge base built upon the KERES ontology. All recorded data are structured and linked with other data in the ontology for further processing.

The goal is to derive reliable recommendations for actions and measurements for protection, avoidance of damage situations, and adaptation. These recommendations will serve as a general instruction basis for actions for decision support on developing individual strategies for climate adaptation for historic gardens and parks.

### 3.4.2. *Franconian Open-Air Museum* in Bad Windsheim, Bavaria

The *Franconian Open-Air Museum* (*Fränkisches Freilandmuseum Bad Windsheim)* is located in the Swabian-Franconian Scarpland (*Schwäbisch-Fränkisches Schichtstufenland)* in the basin between southern Steigerwald and northern Franconian Heights in the south German state of Bavaria. As a result of its location in the basin, the region's climate is rather continental, with hot and dry summers and relatively low rainfall.

The Franconian Open-Air Museum was built starting in 1978 and opened in 1982. It represents the historic Franconian building and cultural landscape, divided into three regional and four thematic groups of construction, covering the period from the early 14th to the second half of the 20th century. The buildings were relocated equally in individual and whole wall parts. Especially the buildings of the late Middle Ages and the inclusion of an urban building group with in situ facilities form a unique feature of the museum.

The now more than 137 buildings of various formats and materials make up the museum one of the largest of its kind in Central Europe but also cause immense building maintenance. The concept of the museum to present the buildings in their traditional complexity of structure, materiality, and signs of age is increasingly causing problems due to their status as freely weathered architectural specimens. Damage to roofs caused by heavy rain events with gale-force winds, wood infestation by beetles and fungi, as well as the failure of building foundations after long periods of drought, have increased significantly over the past 40 years.

### 3.5. *Workshops and Stakeholder Surveys*

The KERES project has so far featured a number of workshops, user surveys, and other meetings, all that contributed to the mind map and ontology beyond regular /hljour fixes, in total more than 15 workshops dedicated to specific topics and 10 stakeholder meetings. Most notably, the meeting with the German Archaeological Institute (DAI KUL-TURGUTRETTER) [25], the workshop with emergency service professionals (BOS) of the THW, and the workshop on the KERES ontology significantly advanced development of the ontology. Eventually, we managed to win stakeholders from a wide variety of specialist areas as participants, including but not limited to emergency service professionals, cultural asset managers of museums and parks, and climate change and climate change impact researchers.

By interviewing managers of technical facilities and plant safety managers, it turned out that often only those categories of disasters are considered that are well-known from past experience, and only as far as it seems absolutely necessary. Concepts for fire protection and extinguishing, as well as for frugal use of water in historic buildings, are often on a very basic and generic level and not adapted for specific cultural assets.

According to the interviews, only for very few large cultural assets advanced concepts seem to exist, such as for the assets of project partner SPSG, while the vast majority of small cultural assets appear to be hardly prepared. Only very few stakeholders have dealt with risks arising from climate change, while at the same time, many of them have observed effects on their assets. For example, one gardener explained that his site needed mowing more often and earlier.

### 3.6. HERACLES *Ontology*

The KERES project and KERES ontology build on the experience gained, among other things, from the HERACLES project and its HERACLES ontology [2]. The HERACLES ontology helps integrate data for the preservation of CH in the context of climate change. The HERACLES ontology was a first step towards designing, validating, and promoting responsive systems and solutions for effective resilience of CH against climate change effects, considering as mandatory premise a holistic, multidisciplinary approach through the involvement of different experts. Still, this ontology already includes quite a large set of declarations and definitions (Figure 4).

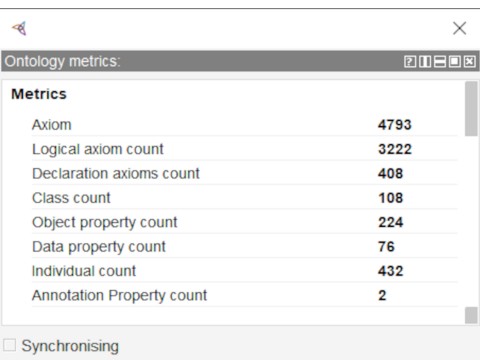

**Figure 4.** The HERACLES ontology as the first step towards CH protection modeling already consists of hundreds of classes, properties, and individuals.

The central elements in the HERACLES ontology are `Cultural Heritage Assets` that need to be protected against `Effects` of climate change. `Risks` arise from climate change effects which can cause `Damage` to CH. A distinction is made between actual `Damage` and `Damage Types` of potential damage. The ontology also models potential `Maintenance Actions` and `Responsive Actions`. To capture climate change relevant parameters, sensors can be modelled following the concepts of the *SensorThings API* standard [26].

Since materials influence how an asset is affected by climate effects in terms of its resilience to weathering and aging, the ontology models information about `Materials` to describe materials and what materials an asset consists of.

*3.7. Mind Map*

The mind map for KERES models a hierarchy of concepts relevant to the project. It emerged from

- prior experience from the HERACLES project,
- all of the workshops, interviews, and surveys (Section 3.5) with domain experts mostly during the early phase of the KERES project (Figure 5),
- but also from external resources such as the *Venice Charter* [27], the *Florence Charter* [28] or the *SiLK guidelines* [29],

Resulting in as many as 569 concepts. While the HERACLES ontology already contains a fair amount of terminology, the mind map extends and elaborates on HERACLES' core terminological areas, such as assets, effects, risks, damages, and actions. Specifically, HERACLES makes barely use of class hierarchies and instead tends to implement entities as flat sets of instances (e.g., class `Material Type` containing instances for specific materials such as `Ancient Mortar`, `Cement Mortar`, `Clay`, `LimeStone`). In contrast, the mind map hierarchically structures these types, for example, as a class hierarchy of materials. Consequently, the hierarchical structure required that we turned HERACLES's instance entities into classes, as we will discuss later (Section 3.8.2). Moreover, the mind map introduces new areas of terminology, such as climate, emergency services, and plants, but also tremendously extends terminology in existing areas, such as adding lots of new asset or damage classes over HERACLES for KERES. Since most discussions and interviews were held in the German language, the mind map focuses on establishing terminology in the German language, while the HERACLES ontology uses English terminology, such that from now on, we have to deal with terminology in two languages. Beyond the hierarchical structure, the mind map does not model any relation between concepts. The task of completing missing relations between concepts (i.e., in terms of ontologies, adding *object properties* ) is subject to the development of the ontology.

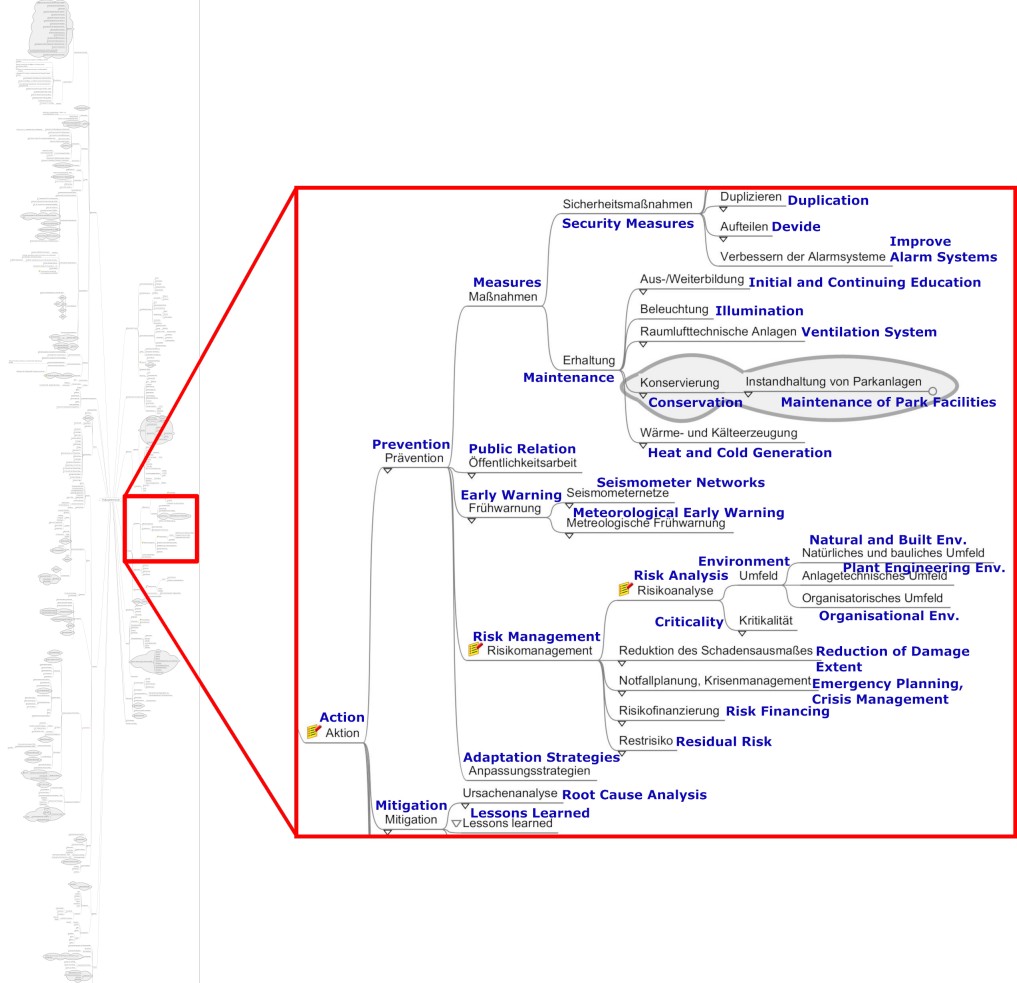

**Figure 5.** In this view of the KERES mind map, detailed comments on concepts are omitted; only concepts themselves are shown, with an enlarged sectional view of the area on preventive actions on the right to give an impression of the overall size of the mind map. Note that the mind map was created solely in the German language; for the convenience of the reader, English labels have been added to the zoomed example excerpt only.

*3.8. KERES Ontology*

Building upon the mind map (Section 3.7) and the HERACLES ontology (Section 3.6), a new ontology tailored for the KERES project requirements was devised [30], going well beyond HERACLES not only in size (Figure 6). In contrast to the HERACLES ontology, the KERES ontology is split into

- a core part with all of the conceptional knowledge and general-purpose individuals that are not expected to change for specific use cases, such as units of measurements, administrative units for specifying locations
- the case studies part that includes individuals in particular of the KERES case studies.

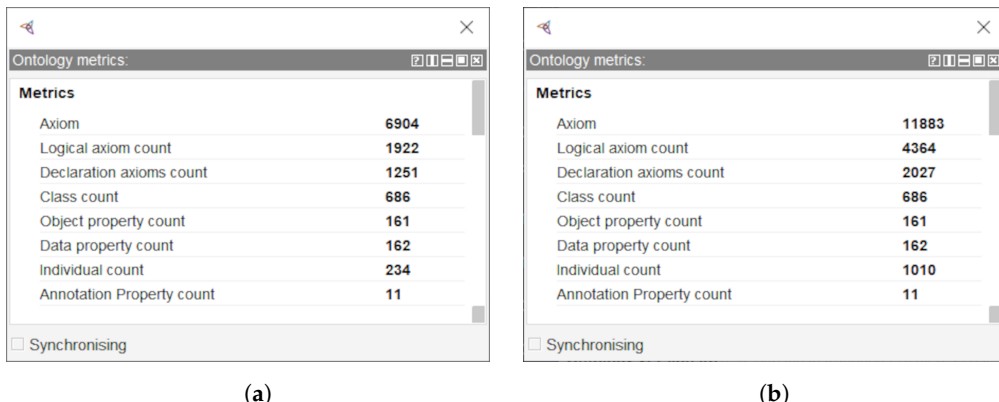

(**a**) (**b**)

**Figure 6.** The size of the KERES ontology goes well beyond that of the HERACLES ontology. (**a**) Already the core part significantly extends on the HERACLES ontology, resulting in as many as 686 classes. (**b**) The complete KERES ontology, including the case studies part, consists of even more individuals.

The core part of the ontology has been published and is freely available as main KERES ontology, hosted on GITHUB [31] and available under the URL https://ontologies.iosb. fraunhofer.de/keres [30], while some sensitive information on the case studies and stakeholders (mostly ABox instances) is encapsulated in the case study part of the KERES ontology.

In the course of developing the KERES ontology, besides the core KERES specific domains of CH, climate, crisis, and related areas, we took the opportunity also to devise and work out some more generic features, including a bookmark concept (Section 3.8.3), ROIs and POIs (Section 3.8.4), and administrative units. We have already factored out all case studies data as a separate ontology module, but all other features are still part of the core KERES ontology. We plan to modularize the ontology further and provide all of these features as separate files for OWL import (Figure 7), such that they can be reused for other projects as well.

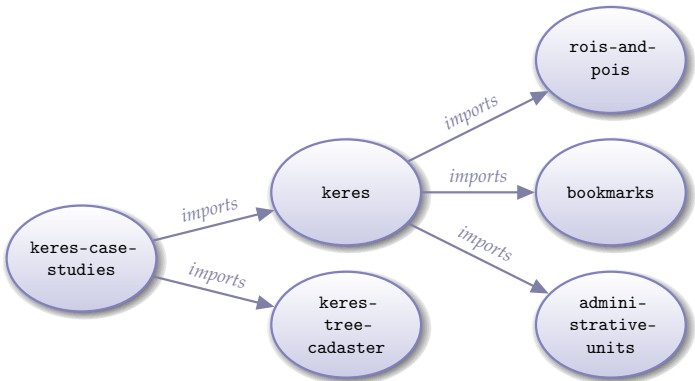

**Figure 7.** The OWL import hierarchy of all ontology modules (work in progress). Module names (even for KERES and ROIs and POIs) are intentionally written in lowercase letters due to module naming conventions.

### 3.8.1. OWL Terminology and Notational Conventions

Since the ontology is implemented in the web ontology language OWL, we adopt OWL terminology and introduce conventions for depicting parts of the ontology.

#### Terminology

The conceptual knowledge, also known as *terminology box* or, shortly, *TBox,* includes OWL *classes* and OWL *properties.* The *facts knowledge,* also known as *assertion box* or, shortly, *ABox,* includes instances of OWL classes and instances of OWL properties.

Notational Conventions

Given the terminology for the aforementioned ontological entities, we now have the building blocks to define notational conventions (Figure 8). Specifically, OWL classes are depicted as circles, OWL object properties, and datatype properties, such as rhombic shapes on light red backgrounds. Instances of classes are drawn as quadratic shapes, and instances of properties as rhombic shapes on light grey backgrounds.

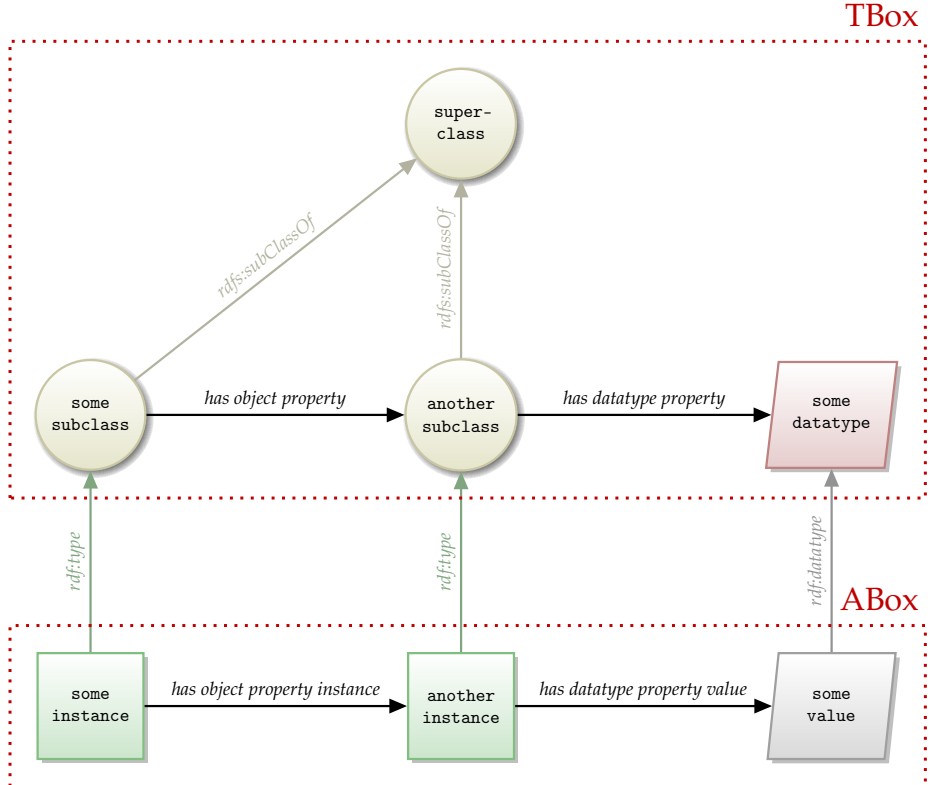

**Figure 8.** Notational conventions for OWL: We depict ontology classes as circles, instances as squares, properties as arrows, and data values as rhombic shapes.

3.8.2. Improvements over HERACLES

While the mind map already anticipates much of the new ontology's hierarchy of classes, it does not consider at all relations between classes (in OWL terms, *object properties)*, nor data to be directly attached to class instances (*dataproperties)*. This is why we also relied on the proven set of properties of the HERACLES ontology (Section 3.6) and tried to re-use them for the new KERES ontology.

Revising HERACLES Ontology

Rather than blindly copying the concepts of the HERACLES ontology, we deployed substantial improvements over HERACLES. First of all, we consolidated the state of the HERACLES ontology with actions for maintenance and refactoring to address minor issues that have been entered into the ontology over time. In particular, we applied the following actions:

- A couple of obvious mistakes (among them, typographical errors, errors in the hierarchy of classes, and assignments of individuals to classes) were fixed.
- Redundant and conceptually overlapping features (presumably from different contributors working on the ontology) have been merged where feasible.
- Since HERACLES uses inconsistent labels and IRI [32] styles (probably another impact of different contributors working on the ontology), we finally decided to devise a set of rules of naming conventions.

- Some poorly chosen terminology has been replaced with better terms, thereby already having in mind the structure of the mind map.

IRI Naming Conventions

IRIs are used in ontologies to identify resources uniquely and can be viewed as further development of URLs and URIs. In contrast to HERACLES, KERES consistently uses the following naming conventions (for clarity, we silently drop the ontology's namespace in the examples):

- IRIs should be human-readable for situations when labels are not immediately accessible nearby the location of the IRI. This rule is useful, for example, for reviewing IRIs in OWL source code, referring to somewhere else.
- Object property IRIs follow the syntactical form
  `<domain-class>_<core-property-name>_<range-class>`, for example:
  `CulturalAsset_isThreatenedBy_Threat` for clarity and avoiding clashes.
- Since the underscore character ("_") is reserved for separating parts of property IRIs, it should not be used as part of class IRIs or for the core property name. More specifically, class names and core property names use Camel Case, preferably just letters and digits, but no minus character ("-").
- Datatype property IRIs follow the syntactical form
  `<domain-class>_<core-property-name>_<range-type>`, for example,
  `District_hasKey_int`. The range type is currently used somewhat more pragmatically rather than strictly formally, and preferably more descriptive, like in
  `ClimateModel_hasAnnualAverageWindSpeed_speed` with `speed` denoting a floating-point value.
- Classes use IRIs in Camel Case form, for example, `CulturalAsset`.

Merge

The next step was to actually merge all concepts of the mind map into the new KERES ontology, thereby following the hierarchy of classes in the mind map. However, the mind map has a structure somewhat differing from HERACLES. In doubt, the KERES structure was preferred over HERACLES since it emerged from the input of a much broader range of stakeholders than were consulted in the HERACLES project.

I18N

While HERACLES was English-only, the KERES project obligates itself to support entity labels and descriptions in multiple languages, supporting at least English and German. Consequently, we put much effort into properly translating terminology relevant to the project from English to German and vice versa. Adding entity descriptions in multiple languages is still an ongoing effort, while we already cover a fair part of all concepts in the ontology.

Property Restrictions

Following the requirements of our applications, most notably those of the WEB-GENESIS server (Section 4.1), the HERACLES ontology mostly complies with OWL DL. In fact, there were very few violations, and all of them have been eliminated while creating the KERES ontology. OWL DL does not allow an entity to be a class and an instance simultaneously, with object properties applying to instances rather than classes. Instead, OWL DL supports *property restrictions* (Figure 9) that can be applied to classes. Unfortunately, enforcing compliance with property restrictions usually requires an OWL reasoner that may not be available or, for performance reasons, can not be applied for each modification, for example, on the ABox of an ontology. HERACLES circumvents these issues by introducing `Type` classes, for example, a class `DamageType` as a complement to the `Damage` class, with `DamageType` containing a set of possible damage type instances that a specific instance of `Damage` can be associated to via an object property `hasDamageType` that relates

specific damage with a specific damage type. An obvious drawback of this approach is that damage types are just a flat set of individuals rather than building a hierarchy. That is, there is no out-of-the-box way in HERACLES to state that, say, corrosion is a specific type of material aging. The mind map, in contrast, sets a hierarchy of damage types that we do want to incorporate into the ontology. Therefore, instead of having two classes `Damage` and `DamageType` as HERACLES does without any further subclasses, in the KERES ontology, we dropped the `DamageType` class and instead model damage types as subclasses of class `Damage` (Figure 10). To express that a specific damage is of specific type *corrosion,* in KERES, we model the specific damage as instance of class `Corrosion`, while in HERACLES, it is modelled as instance of class `Damage` with object property `hasDamage` that links to instance `Corrosion` of class `DamageType`.

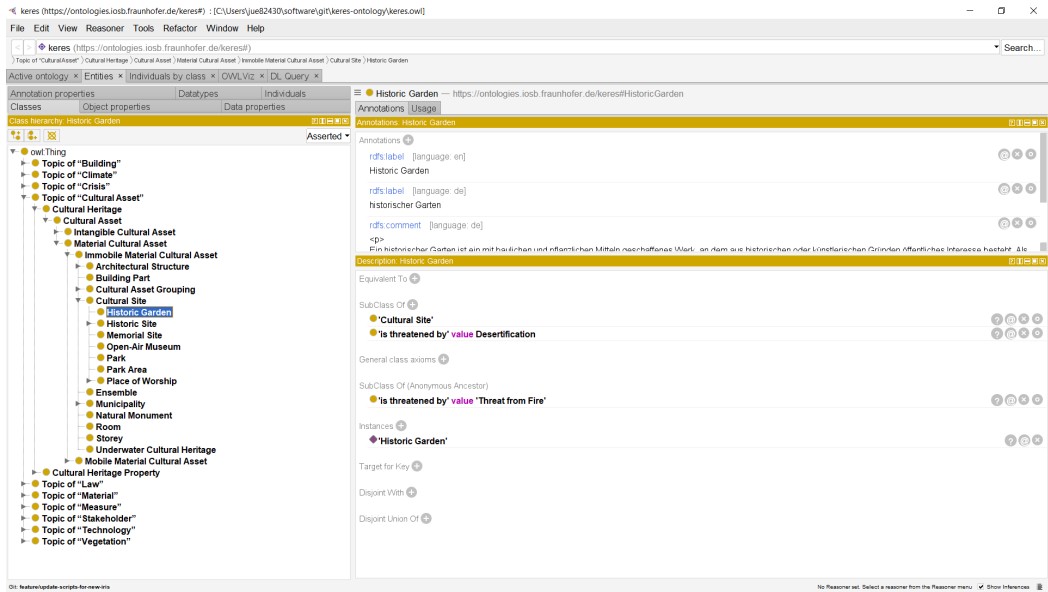

**Figure 9.** Property restrictions of a class as shown in PROTÉGÉ 5.5.0 [33]. With these restrictions, each individual of class `HistoricGarden` is declared to hold property `is threatened by` with values `Desertification` and `Threat from Fire`. The former property value restriction is directly declared on class `HistoricGarden`, while the latter restriction is declared on and derived from a super-class. The screenshot also shows the prototype instance of this class, which differs from (i.e., has a different IRI than) the class itself.

Having a hierarchy of damages in the KERES ontology rather than a flat set of damage types as in the HERACLES ontology has several advantages. Most notably for us, applications built on top of the ontology can enhance browsing of damage types by presenting a tree that follows the class hierarchy rather than just offering a huge, semantically randomly ordered drop-down list of damage types to select from. Moreover, object and datatype properties can be restricted to specific damage classes within the class hierarchy rather than flooding the top-level `Damage` class.

However, an issue remains: How can we express that a specific damage type is associated with a specific individual or data value? In HERACLES, for example, the `Corrosion` instance of the `DamageType` class is assigned property `isCausedByEffectType` that links to the `EffectType` *air pollution,* to express that air pollution can contribute to corrosion (Figure 11). In contrast, in the KERES ontology, we have instead a class `AirPollution` that is a subclass in the class hierarchy beneath the top-level class `Threat` (Figure 12). How can we express a statement corresponding to that in HERACLES? First of all, we provide a property restriction on class `AirPollution` for applications that are aware of OWL property restrictions. For applications that do not recognize property restrictions, we additionally define a *prototype individual* that is an instance of class `AirPollution`. Similarly, we define a prototype individual of class `Corrosion`. Given these two individuals,

we now can introduce an ordinary object property `canCauseDamage` that links from domain class `Threat` to range class `Damage` and apply it for these two prototype individuals to express that threat `air pollution` can cause damage `corrosion`.

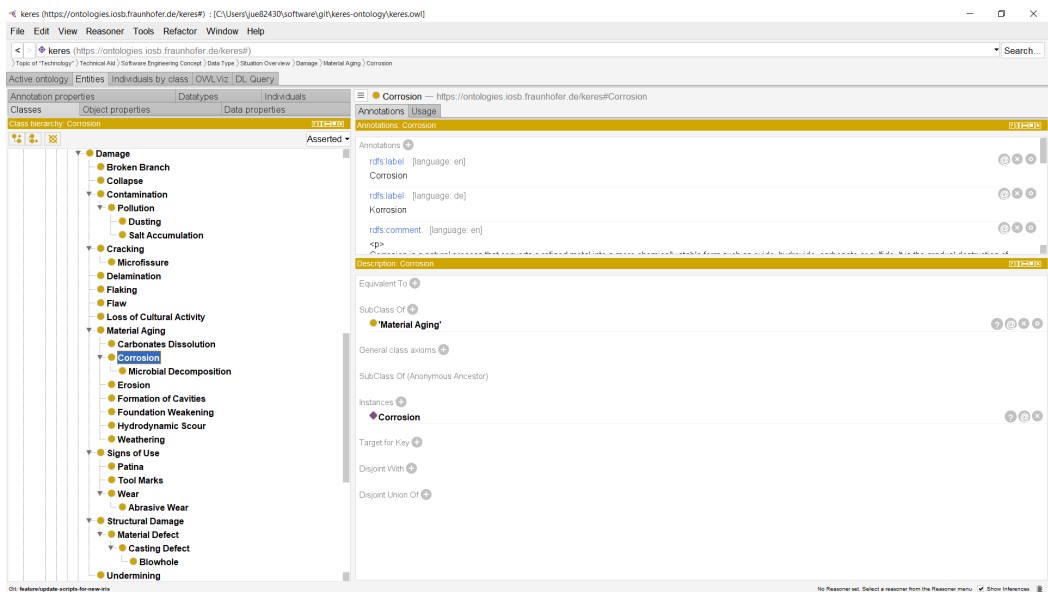

**Figure 10.** In KERES, damage types are modelled as subclasses of the `Damage` class rather than having another class `DamageType` as in HERACLES, that contains all types as flat set of individuals.

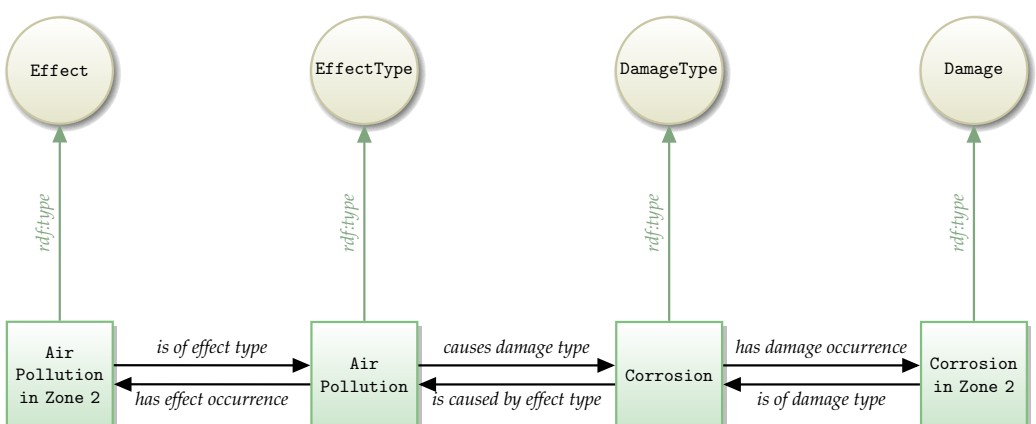

**Figure 11.** Threats (≅ effects) and damages in HERACLES.

### 3.8.3. Bookmarks

Some applications building upon the KERES ontology require a selected list of instances. For example, the WEBGENESIS web application (Section 4.1) features a web page that lists cultural assets of all KERES case studies in a dashboard-like manner. Similarly, another application uses that same list of bookmarks as starting point for finding measurements for some specific cultural asset. Yet another specific bookmark list has been created for an application that presents all assets available for reference when creating a route card.

Technically, membership of an object instance in a list of bookmarks could be modeled as Boolean datatype property on the bookmarked instance itself, thus, marking whether the instance is part of a specific bookmark list or not. However, for each new list of bookmarks, we would have to introduce yet another Boolean property for *all* objects that *could* be part of that bookmark list, thus, polluting lots of objects with a huge number of Boolean datatype properties that most of them will never use. Moreover, beyond a simple Boolean marker of membership, we may like to model additional information such as order (*rank*) within the list of bookmarks, or maybe attach additional descriptive

text. Finally, a typical characteristic of bookmarks is their highly user-specific application, and, thus, conceptionally belongs to the bookmarking user rather than to the bookmarked object instance. Note that, while technically possible, it would be a misachievement to model membership of objects in a bookmark list as class membership of that list since, conceptionally, a bookmarked entry is not an instance of a bookmark list.

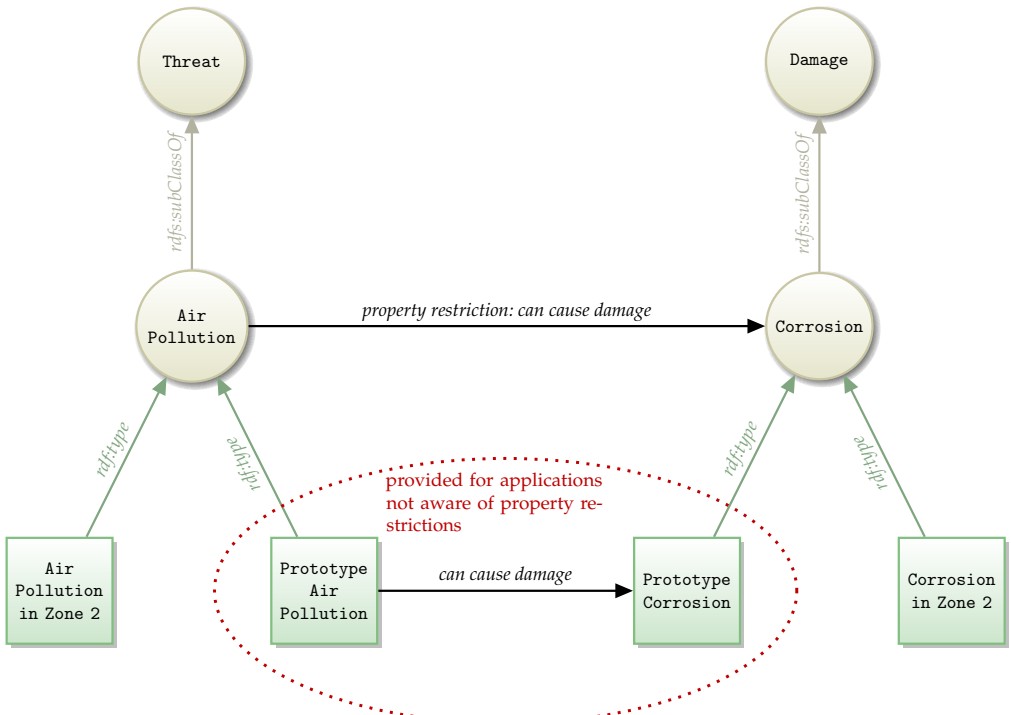

**Figure 12.** Threats and damages in KERES.

Hence, rather than modeling bookmarks via class membership or Boolean datatype properties on the bookmarked objects themselves, the KERES ontology instead introduces new concepts `BookmarkEntry` and `BookmarkFolder` (Figure 13). A `BookmarkFolder` instance may be linked to any number of `BookmarkEntry` instances or nested `BookmarkFolders`. In fact, `BookmarkFolder` is just a special kind of `BookmarkEntry` and, thus, declared as a subclass of `BookmarkEntry`.

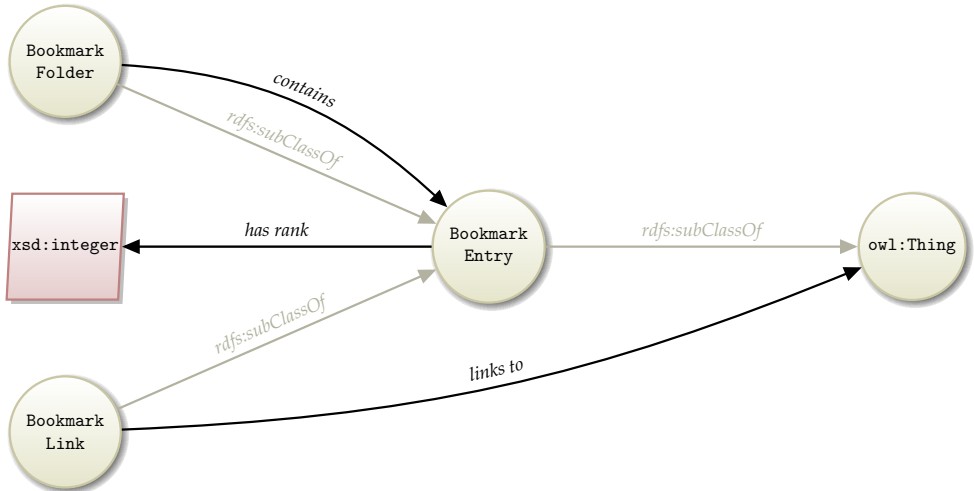

**Figure 13.** Bookmarks in KERES.

The order of appearance of bookmark entries within a bookmark folder can be controlled by a *rank* value that is attached as an integer datatype property to each `BookmarkEntry`.

We prefer to use a rank value over modeling bookmark entries as a linearly chained list of object entities since in SPARQL, we can retrieve all entries of a bookmark folder by simply querying for all entries linked to a bookmark folder and sorting them with a simple `ORDER BY ?rank` clause. With a chained list, retrieval of all entries in the correct order would require much more effort.

Each `BookmarkEntry` instance links to the actual object instance that it refers to. This object instance can be any instance available in the ontology; hence, we choose `owl:Thing` as a type for bookmarked objects.

In the context of KERES, we use bookmarks for several purposes:

- The bookmark folder `KERES Case Studies` links to each one of the five case studies (Section 3.4). This list of links is used, for example, by the KERES knowledge base server that depicts all of the case studies on a clickable map (Figure 3). Adding just another case study to this list of bookmarks is sufficient for this new case study to automatically appear on the map as well with a proper link, which is also extracted from the ontology.
- Bookmark folder `EU OMC Best Practice Examples` links to each of 83 best practice examples' short information that has been collected in the course of the KERES project. As of now, the knowledge base server does not make specific use of this list; still, this list of bookmarks is accessible via the server's SPARQL (cp. Figure 14) interface, such that external applications can make use of the bookmarks as well.
- Finally, bookmark folder `PSE Assets` is used for managing a list of cultural assets accessible for the external application WALKER for creating and managing route cards (Section 4.3).

```
1  PREFIX rdfs: <http://www.w3.org/2000/01/rdf-schema#>
2  PREFIX dc: <http://purl.org/dc/elements/1.1/>
3  PREFIX keres: <https://ontologies.iosb.fraunhofer.de/keres#>
4  BASE <https://ontologies.iosb.fraunhofer.de/keres-case-studies>
5
6  SELECT
7    ?caseStudy ?label ?country ?title ?description
8  WHERE {
9    <#BookmarkFolder_EU-OMC-Best-Practice-Examples>
10       keres:BookmarkFolder_contains_BookmarkEntry ?bookmarkEntry .
11   ?bookmarkEntry keres:SpecificBookmark_linksTo_Thing ?caseStudy .
12   ?bookmarkEntry keres:BookmarkEntry_hasRank_rank ?rank .
13   ?caseStudy rdfs:label ?label .
14   ?caseStudy dc:title ?title .
15   ?caseStudy dc:description ?description .
16   ?caseStudy keres:Action_isPerformedBy_Stakeholder ?country
17   FILTER(?country = keres:Country_Germany) .
18 }
19 ORDER BY ASC(?rank)
```

**Figure 14.** A SPARQL query for retrieving all EU OMC Best Practice Examples located in Germany, following the original order of publication.

We expect the need for more bookmark lists to appear in the near future.

### 3.8.4. ROIs and POIs

A *region of interest* (*ROI*) is a widely and commonly used concept for formally identifying a region for a particular purpose. A region of interest may contain any number of *points of interest* (*POI*).

Originally, ROIs were introduced in KERES for modeling regions that contain cultural assets of interest. A POI in KERES is simply a geolocation usually within an ROI, for example, for specifying the location of a particular artefact (Figure 15).

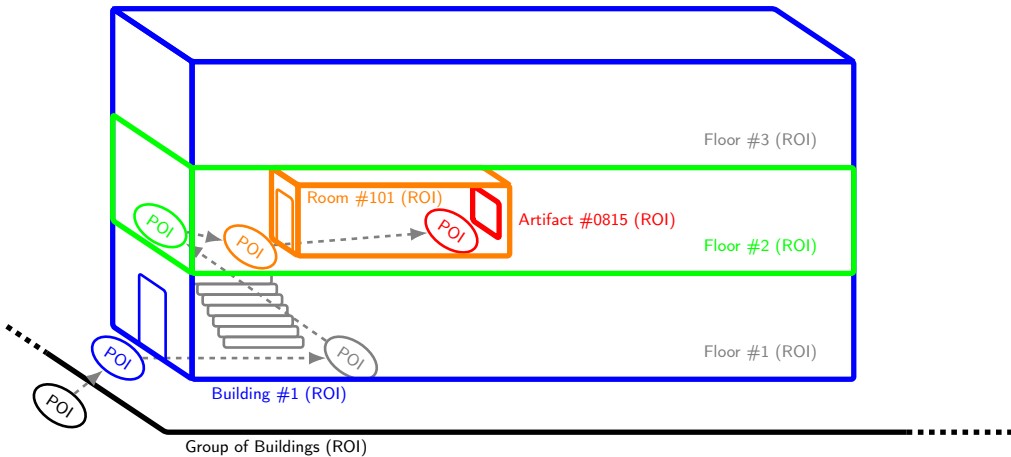

**Figure 15.** Regions of interest (ROIs) are hierarchically nested and, thus, form a tree-like structure, for example representing a group of buildings, the individual buildings, floors, rooms, and artefacts within the rooms. Points of interest (POIs) are located within ROIs and can be interlinked to form a directed graph for modeling paths even across ROIs.

ROIs are nested via the `is located in` object property. For example, the ROI representing a building as a cultural asset will typically contain ROIs that represent the floors of that building. Each ROI that represents a floor will typically contain a number of ROIs that represent the rooms and corridors within that floor. Likewise, a single room may contain ROIs that divide the room itself into even smaller regions, for example, a showcase with cultural artifacts, and so forth (Figure 16). Each ROI has an extent (`BoundingBox`) and, if applicable, a position relative to its parent (`RelativePosition`).

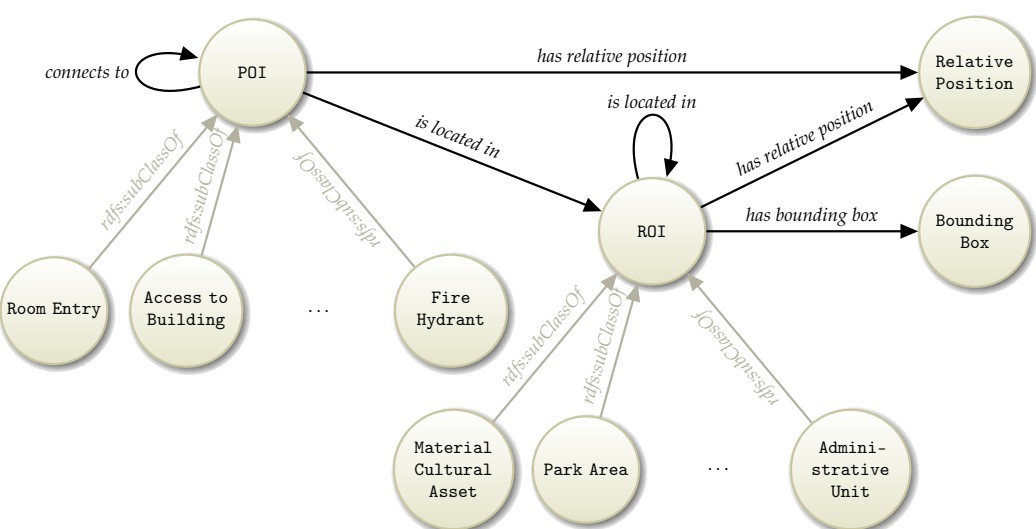

**Figure 16.** ROIs and POIs in the KERES ontology.

In contrast, POIs are not nested. Instead, each POI is usually located in a single ROI, which, of course, itself may be located within another ROI, etc. That is, POIs can be thought of as leaves of a tree of ROIs. Furthermore, POIs can be interconnected via the `connects to a` object property, even if they are part of different ROIs. A sequence of interconnected POIs defines a *directed path.* Paths may branch and join, thus, effectively creating a *directed graph* with the POIs taking the role of graph nodes. A path may, for example, describe a tour through a building. Just like ROIs, POIs have a position relative to the ROI they are located in.

While ROIs and POIs were originally added to the KERES ontology for future use in AR/VR applications, such as virtual or augmented museum tours across cultural assets

(Figure 17), they also turned out to be a useful tool in various other applications, including, for example, administrative units or for describing relations in more complex ensembles of CH, including a subdivision of parks into park areas in the tree cadaster (Section 3.8.6).

```
1  PREFIX keres: <https://ontologies.iosb.fraunhofer.de/keres#>
2  BASE <https://ontologies.iosb.fraunhofer.de/keres-case-studies>
3
4  SELECT DISTINCT ?asset ?assetRank (GROUP_CONCAT(?roi ; separator=' ; ') AS ?rois)
5  WHERE {
6    <#BookmarkFolder_PSEAssets> keres:BookmarkFolder_contains_BookmarkEntry
7        ?bookmarkEntry .
8    ?bookmarkEntry keres:SpecificBookmark_linksTo_Thing ?asset .
9    ?bookmarkEntry keres:BookmarkEntry_hasRank_rank ?assetRank .
10   { ?roi keres:ROI_isLocatedIn_ROI+ ?asset }
11 }
12 GROUP BY ?asset ?assetRank ORDER BY ASC(?assetRank)
```

**Figure 17.** A SPARQL query for retrieving all ROIs within all assets listed in a specific bookmark folder. The ROIs are grouped by asset.

### 3.8.5. Topics

When creating the KERES ontology, the top-level folder of the class hierarchy initially grew very quickly and became confusingly large. We decided to group the top-level classes into *topics* that now make a handy set of global entry points into the ontology. The topics are (in alphabetical order):

- Topic of "Building";
- Topic of "Climate";
- Topic of "Crisis";
- Topic of "Cultural Asset";
- Topic of "Law";
- Topic of "Material";
- Topic of "Measure";
- Topic of "Stakeholder";
- Topic of "Technology";
- Topic of "Vegetation".

Given these topics as new top-level classes, we re-classified all of the previous top-level classes as subclasses. Remember that the hierarchy of classes is not only relevant to developers but also visible on the WEBGENESIS Server (Section 4.1) to end users for browsing through a corresponding tree of web pages. These topics now appear as top-level web pages and help end users quickly find relevant bits of the ontology.

### 3.8.6. Tree Cadaster

The KERES case studies include not only buildings as cultural assets but also natural monuments like parks. The overall health of a park can be qualified best by the vitality of its plants. For this purpose, the KERES ontology features a *tree cadaster* (Figure 18) that, for each single recorded tree, contains an object instance in the ontology. Attached to each tree are, as properties, a unique identifier, its top diameter, and its function by one out of 10 possible categories (among them, *avenue tree*, *big bush*, *dominant*, or *frame tree)*, as well as its vitality by one of 5 quantitative values (*exploration, degeneration, stagnation, resignation, dead)*, wood species, and, optionally, a link to a photo, but also information on substrate additives and their felling date, if applicable.

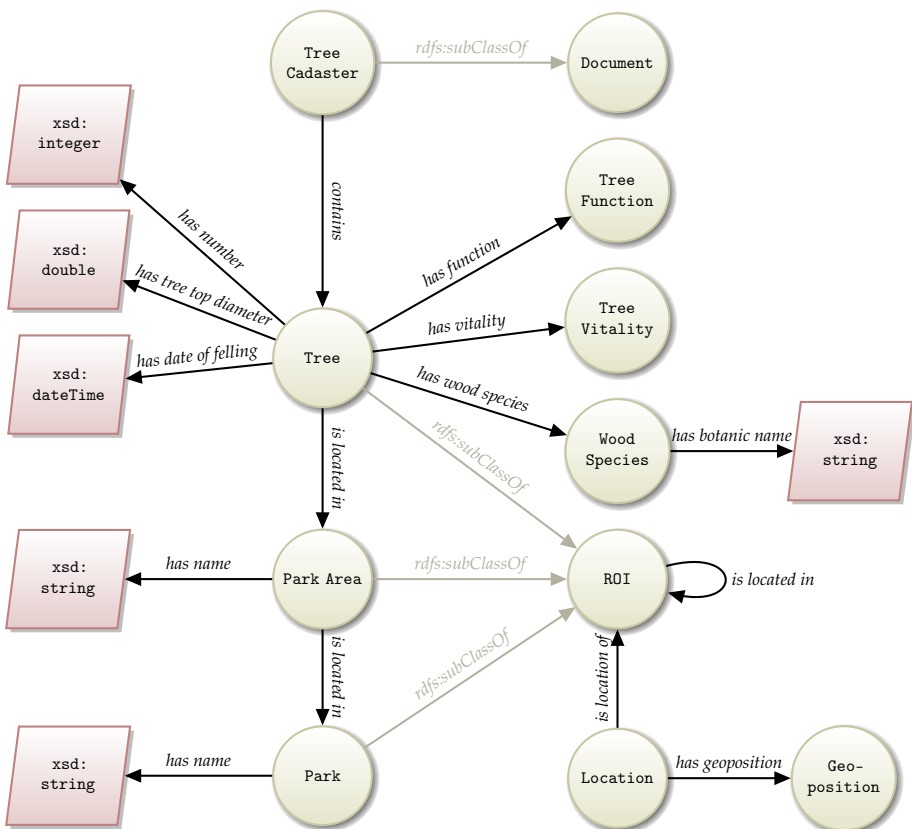

**Figure 18.** Tree cadaster in the KERES ontology. For clarity, only a selected small set of datatype properties is shown, and some intermediate subclasses have been omitted.

Once more, ROIs (Section 3.8.4) turned out to be useful, in this case, for mapping each tree to its park area and each park area to its enclosing park, but also for providing geospatial data in a standard manner as for all of the ROIs. Tree function, vitality, and wood species are modeled as classes `TreeFunction`, `TreeVitality`, and `WoodSpecies`, respectively, with a predefined set of possible instances to choose from.

Unfortunately, the tree cadaster inflates the size of the ontology. It turns out that when integrating all the tree data provided by project partner SPSG into the ontology, the cadaster accounts for more than 99% of the size of the ontology. More than 80,000 trees are listed in the cadaster, so the size of the OWL code is multiplied from around 2 MB to around 290 MB.

As a consequence, we factored out the tree instances data of the KERES ontology as a module, exploiting the linked data [34] capabilities of OWL, and put this module onto a separately running SPARQL server backed by a FUSEKI [35] instance.

With the ontological model of the tree cadaster, one may easily run evaluations on the cadaster with simple SPARQL queries, for example, querying for all felled trees to be grouped by the year of felling (Figure 19).

### 3.8.7. EU OMC Best Practice Examples

The EU *Open Method of Coordination* (OMC) [36] group of Member States' experts has collected a total of 83 best practice examples for protection of CH with contributions from 26 member states [37]. The goal of these examples is to

- exchange policies, foreseen threats or impacts, and proposing strategies and innovative measures to avoid or reduce the climate impact on CH, including early warning systems, and
- exchange on the response of CH sites and institutions and communities to mitigate impacts of climate change on CH in accordance with the European Green Deal.

```
1  PREFIX rdfs: <http://www.w3.org/2000/01/rdf-schema#>
2  PREFIX rdf: <http://www.w3.org/1999/02/22-rdf-syntax-ns#>
3  PREFIX keres: <https://ontologies.iosb.fraunhofer.de/keres#>
4
5  SELECT (COUNT(?tree) AS ?count) (SAMPLE(?fellingYear) AS ?yearOfFelling)
6  WHERE {
7    ?tree rdf:type ?treeType .
8    ?treeType rdfs:subClassOf* keres:Tree .
9    ?tree keres:Tree_hasDateOfFelling_date ?fellingDate .
10   BIND (year(?fellingDate) AS ?fellingYear)
11 }
12 GROUP BY ?fellingYear ORDER BY ASC(?fellingYear)
```

**Figure 19.** A SPARQL query applied on the tree cadaster ontology for querying felled trees and grouping them by the year of their felling date.

The KERES ontology introduces a concept `Case Study` (Figure 20) for incorporating title, description, and responsible `Stakeholder` (i.e., the member state, modeled as `Administrative Unit`) for each of the 83 best practice examples, collected in a dedicated `Bookmark Folder` (Section 3.8.3). For storing project titles and descriptions, we prefer to use the more specific Dublin Core [38] datatype properties `dc:title` and `dc:description` over the `rdfs:label` datatype property, since the latter has very broad and generic usage, that often leads to misuse and unclear semantics. Thanks to the bookmarks concept, users can either use the WEBGENESIS knowledge base server's standard navigation for browsing through the collection of OMC case studies via entries in the bookmark folder or a (yet to be implemented) application or plug-in for direct navigation similar to the map of the KERES case studies (Section 3.4).

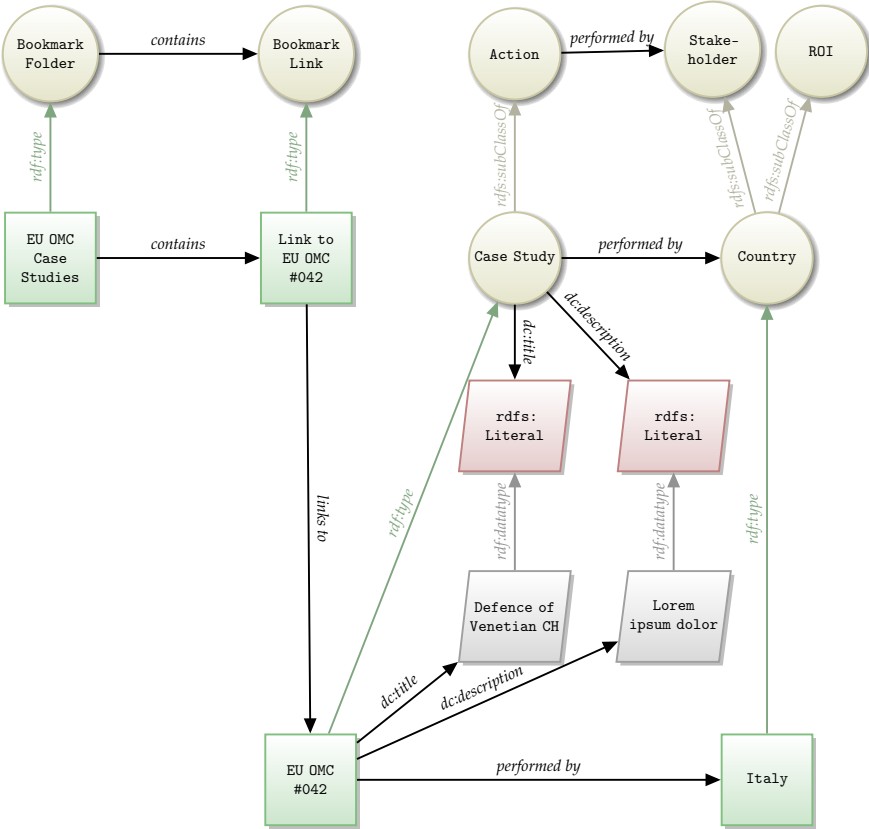

**Figure 20.** The `Case Study` class in the KERES ontology is a subclass of `Action`, thus, deriving the object property that connects to the `Stakeholder` class, here with case study *EU OMC #042* performed by stakeholder *Italy*. The *Lorem ipsum dolor* value is a placeholder for an actual descriptive text on the case study. Some intermediate subclasses are omitted for clarity.

## 4. Results

For verification and evaluation of practical benefits of the KERES ontology, we deployed it in the WEBGENESIS server (Section 4.1), but also successfully implemented small applications that are built upon the ontology. Moreover, dozens of SPARQL queries have been used to test correctness and usefulness of the ontological model.

### 4.1. WEBGENESIS

WEBGENESIS is a web application for information management and information retrieval with ontology support. For an imported OWL ontology, such as the KERES ontology, it provides a tree-structured set of web pages representing the ontology's class hierarchy. Similarly, WEBGENESIS represents each instance entity of the OWL ontology as a separate web page of its own, as well as each datatype property and object property. These pages are interlinked as appropriate; for example, the page that represents an OWL class contains links to the pages that represent instances of this class. Similarly, WEBGENESIS interlinks classes and object/datatype properties. For data entry of new instances, WEBGENESIS automatically creates pages with proper HTML input forms for datatype and object properties. It is even possible to add, modify or delete classes and datatype and object properties.

All of these pages can be individually customized (Figure 21). By default, `rdfs:comment` entries, if available for the corresponding OWL entities, are displayed as description text on the pages. However, WEBGENESIS supports even further customization, such as overriding the default descriptions, customizing HTML input forms, and even modifying the layout of entire individual pages.

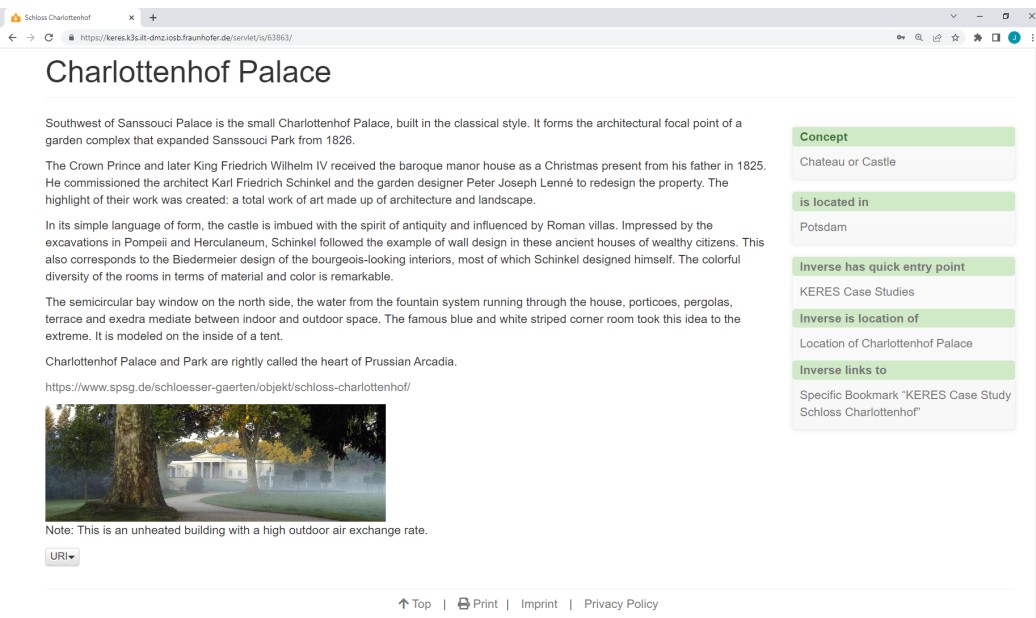

**Figure 21.** For beneficial browsing through an ontology, WEBGENESIS displays `rdfs:comment` and `rdfs:label` information of the ontology OWL file, such that ontology entities (in this example, an instance of OWL class `Castle`) are rendered in a neat way (screenshot of web browser view).

Besides a standard keyword-based search, WEBGENESIS also supports information retrieval based on SPARQL queries that are executed on the ontologies hosted on a WEBGENESIS instance. In summary, WEBGENESIS features a fully-fledged web-based information management system with seamlessly integrated support for OWL ontologies.

In the course of KERES, we deployed the KERES ontology on WEBGENESIS, but also customized many pages to gain optimal user experience for the use of WEBGENESIS as database management and information retrieval system tailored for KERES. For example, on the landing page (Figure 22), we integrated a dashboard that links to the KERES case

studies, to the KERES FINDER (Section 4.2), and more items. For configuring case study entries, we exploit—once again—the ontology's bookmark feature (Section 3.8.3), with each bookmark linking to an entity in the ontology that describes the corresponding case study, including its name, an icon, and its geographical location. Using a standard geodata service, WEBGENESIS generates a clickable geographical map (Figure 3) that, for each case study asset, shows its location and surrounding municipality.

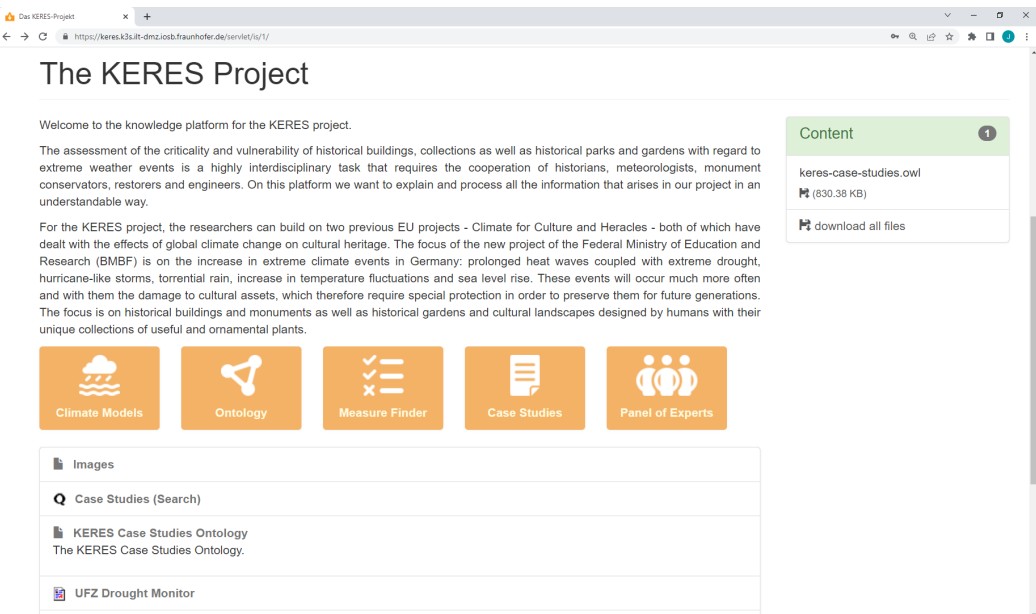

**Figure 22.** The landing page links to climate models, a browsable view of the ontology, the KERES FINDER, the case studies, and the board of experts (screenshot of web browser view).

### 4.2. KERES FINDER

KERES FINDER is an extension feature for the WEBGENESIS web server application for CH managers exploring actions against possible threats to their CH. KERES FINDER considers previously collected data stored in the KERES ontology for analyzing possible threats and suggesting proper actions. Implemented as a wizard within WEBGENESIS, it prompts the user with a sequence of questions on some specific CH and its environment. Based on the user's input, KERES FINDER guides the user towards specific suggestions for protecting the CH in question. The application makes heavy use of the ontology by traversing `Threat`, `Damage`, `Action` and `Cultural Asset` and instances of derived classes as needed, thus, proving the practical benefit of the KERES ontology.

### 4.3. WALKER—*Route Card Managing*

WALKER is a web application that we developed to ease the creation of route cards for helping professional firefighters in rescuing mobile CH in the case of an emergency. The structure of route cards follows a concept developed by the *Bayerische Schlösserverwaltung (BSV)* [39] in cooperation with the emergency service staff of the city of Munich. WALKER is an external application that makes use of WEBGENESIS's REST API for gaining access to the KERES ontology. WALKER makes heavy use of concepts and individuals of the KERES ontology, particularly those related to cultural assets, persons and stakeholders, ROIs, and links to external resources. For example, whenever possible, the applications look up available instances, such as institutions and assets, for creating drop-down lists, rather than the user having to type in data. In fact, from implementing this application, we learned to know that WEBGENESIS's REST API was missing a registry for looking up ontology endpoints, which we consequently upgraded in WEBGENESIS.

Once again, ROIs (Section 3.8.4) proved a helpful tool, here for localizing cultural assets. Specifically, ROIs help identify those municipalities that will appear in the drop-

down list of WALKER's municipality selection field. Moreover, we expect POIs to prove useful tools for describing routes through an asset—actually, the core purpose of a route card. Though, as of now, creating route card maps is not yet the subject of WALKER.

## 5. Discussion

Building upon previous work such as the HERACLES project, with updated and new input from domain experts of areas relevant for the KERES project, like CH protection and climate change, and work on all of the five case studies in close cooperation with the project partners, the KERES ontology successfully addresses and supports all central issues and challenges exposed by the project. Integrated into the KERES server, the ontological knowledge is directly accessible via browsing through the ontology and by performing SPARQL queries. Moreover, we provide dedicated applications for exploiting knowledge from the database for specific tasks, such as WALKER or the KERES FINDER, thus, showing the practicability and usefulness of the ontology. Yet, there is still room for improving the usefulness of specific query results.

### 5.1. ROIs and POIs

Originally not foreseen, the desire for supporting regions and points of interest came up when a project partner asked for the possibility of describing the structure of a cultural asset as well as paths through an asset, for example, for guided tours. Very soon, it turned out that ROIs are also useful in the context of WALKER to describe parts of assets and emergency ways for rescue services. Similarly, ROIs proved useful for modeling administrative units and, shortly afterward, for modeling park areas within parks in the tree cadaster. We are confident that we will find even more applications for the concept of ROIs and POIs.

### 5.2. Bookmarks

Similarly, the concept of bookmarks also turned out to be a versatile feature. Originally designed as a flat list of links for a dashboard, we soon extended it also to cover any hierarchy of bookmarks and applied it for the list of the five KERES case studies used, for example, by the KERES FINDER, and also for a set of cultural assets to start within the WALKER application, and for the list of EU OMC best practice examples. Again, we are confident to find even more applications.

### 5.3. Quality of Query Results

While applications WALKER and KERES FINDER basically deliver adequate results for queries, we have observed two limiting factors regarding their quality:

- the amount of instance data modeled in the ontology
- the ranking of query results.

First of all, query results encompass, of course, only instances that have been modeled in the ontology. Currently, the KERES ontology without tree cadaster consists of only some hundred instances, and, more specifically, only some dozens of threats and prevention measures, that we identified mostly in the course of the interviews with domain experts and while working on the case studies in close cooperation with the project partners. Obviously, the more instances are modeled in the ontology, the more likely a query delivers highly valuable results. That is, in the long term, we strive to get much more data into the knowledge base.

Secondly, ranking of query results may turn out to be crucial, especially when the amount of data modeled will grow over time. Already now, without any further precautions, for example, the KERES FINDER suggests a fairly large number of preventive measures. To limit the number of results, we currently let the user further narrow the query, for example, by choosing among possible threats computed in one step before computing possible preventive measures in the next step.

In the long term, we envision ranking threats and measures from the probability that a threat will actually occur, based, for example, on evaluating risk matrices or even more sophisticated models like process chains.

**Author Contributions:** Conceptualization, J.M.; methodology, J.M., J.R. and T.H.; software, T.H. and J.R.; validation, J.R. and T.H.; investigation, J.M. and J.R. and T.H.; data curation, J.R.; writing—original draft preparation, J.R.; writing—review and editing, J.M., T.H. and J.R. and P.H.; visualization, T.H. and J.R.; supervision, J.M. and P.H.; project administration, funding acquisition, All authors have read and agreed to the published version of the manuscript.

**Funding:** This research was carried out as part of the KERES project funded by the German Federal Ministry of Education and Research, grant number 13N15443.

**Institutional Review Board Statement:** Not applicable.

**Informed Consent Statement:** Not applicable.

**Data Availability Statement:** Data are available under the terms of Creative Commons Attribution 4.0 International License (CC-BY 4.0). The KERES ontology is publically available at https://ontologies.iosb.fraunhofer.de/keres.

**Acknowledgments:** We would like to thank all experts from all of the case study project partners as well as all external experts for their highly valuable input in developing the ontology and related applications. The KERES tree cadaster is founded on the SPSG's data structures for their tree cadaster GEOSPARQL server, which we have adopted for use in the KERES ontology. The concept of ROIs and POIs emerged from cooperation with Fraunhofer Heinrich-Hertz-Institute from their visualization for CH project as part of FALKE II [40].

**Conflicts of Interest:** The authors declare no conflict of interest.

## Abbreviations

The following abbreviations are used in this manuscript:

| | |
|---|---|
| API | Application Programming Interface |
| CH | Cultural Heritage |
| HERACLES | HEritage Resilience Against CLimate Events on Site |
| KERES | Kulturgüter vor Extremklimaereignissen schützen und RESilienz erhöhen |
| OWL | Web Ontology Language |
| POI | Point of Interest |
| ROI | Region of Interest |
| SPARQL | SPARQL Protocol and RDF Query Language |
| SPSG | Stiftung Preußische Schlösser und Gärten Berlin-Brandenburg |
| THW | Bundesanstalt Technisches Hilfswerk |

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
