# Peer review of "The KERES Ontology: Protecting Cultural Heritage from Extreme Climate Events"

_heritage, doi:10.3390/heritage6050211_

Round 1
Reviewer 1 Report
This manuscript is presenting the KERES project that is the successor of a previous project -the HERACLES one- that delt with the recovery of monuments/cultural heritage affected by climate change. As far as I am concerned, the HERCULES project was quite unique and I am happy that its objectives are still applied and expanded, through the KERES project.
From the above-mentioned, there is no doubt that the proposed manuscript is within the scope of Heritage. The manuscript is well written and well presented. Some minor remarks to be taken into consideration before its acceptance, if applicable:
1. 1. Please, consider explaining the KERES abbreviation and the basics of the project (that are now in lines 186-190) and also the scope of the previous HERACLES project in the beginning of the introduction. I strongly believe that many potential readers will not be familiar with both projects.
2. 2. Figure 4. It is clear that a lot of work has been done in order to support multiple languages through the KERES project, but the present manuscript will be addressed to potential audience that may be not familiar with German. Please, present it with nominations inside the figure in English. I’m referring to the title inside the figure (Die KERES-Fallstudien) and to the text on the left (Die KERES-Fallstudien sind … den einzelnen Studien). Can the resolution of the print screen (map) be improved?
3. 3. Figure 8. The same as with the previous comment.
4. 4. Figure 9. Some refinements in the word-object, such as “keres” and “rois” in uppercase etc.
Author Response
> 1. Please, consider explaining the KERES abbreviation and the basics of the project (that are now in lines 186-190) and also the scope of the previous HERACLES project in the beginning of the introduction. I strongly believe that many potential readers will not be familiar with both projects.
Done: The text has been moved to the introduction. The scope of HERACLES has been added.
> 2. Figure 4. It is clear that a lot of work has been done in order to support multiple languages through the KERES project, but the present manuscript will be addressed to potential audience that may be not familiar with German. Please, present it with nominations inside the figure in English. I’m referring to the title inside the figure (Die KERES-Fallstudien) and to the text on the left (Die KERES-Fallstudien sind … den einzelnen Studien). Can the resolution of the print screen (map) be improved?
Done: The screenshot has been replaced with an English version of the web page. The resolution should be better now.
> 3. Figure 8. The same as with the previous comment.
Since Fig. 8 of the original submission does not contain German text, instead Fig. 7 is assumed to be meant. Fig. 7 shows an excerpt of the mind map. There is no English version of the mind map (it is the ontology itself that introduced support for multiple languages, not the mind map). For benefit of the reader, there are now additional English labels added to Fig. 7 that translate the depicted concepts. The caption has been extended to hopefully make clear that these labels are not part of the mind map, but have been added only in this figure. Additionally, the screenshots of Fig. 23 and Fig. 24 have been replaced with corresponding English versions.
> 4. Figure 9. Some refinements in the word-object, such as “keres” and “rois” in uppercase etc.
“keres” (instead of “KERES”) and “rois” (instead of “ROIs”) are intentionally used in the figure, since these identifiers represent the module names (effectively, file names without suffix), for which a lowercase convention holds. The caption of the figure has been extended to hopefully make clear why lowercase is used in this figure.
Reviewer 2 Report
l.26: “API” Probably more, or less, known abbreviation to many in CH, but could be spelled out first time used.
l.53: “…as a mind map.
l.55: “The main focus…”
l.317-318: “...for recording information on materials and of which materials an asset consists of.”
l.322: “..during the early phase..”
Figure 6: This figure seems superfluous; it only gives the number of nodes = 569 concepts as (could be better if needed) described in the text. (the “leaf nodes” and “branches” are not mentioned in the text, but could be without the figure if needed)
l.357: “…as a separate ontology module,..”
Chap 3.8.1. It may be that the first paragraph in this section belongs to the previous section and the last paragraph to the next section, before Fig. 10 ?
l.430: Were “few of the violations eliminated”, or “were there few violations and they have been eliminated” ?
l.446, 463: Figure notations (Fig 13) and (Fig 14) are in different subsections than the figures which is unfortunate for the reading. This is also the case with other figures. If possible, figures should be inserted where noted in txt. (Some journals change figure placement to first figure notation in the text in the final editing..)
Figure 17: The fig. should probably not be inside the topics list (?)
Author Response
> l.26: “API” Probably more, or less, known abbreviation to many in CH, but could be spelled out first time used.
The abbreviation is now introduced the first time it is used.
> l.53: “…as a mind map.
An article has been added.
> l.55: “The main focus…”
An article has been added.
> l.317-318: “...for recording information on materials and of which materials an asset consists of.”
Done: The text has been rephrased as “for the purpose of describing materials and what materials an asset consists of.”
> l.322: “..during the early phase..”
An article has been added.
> Figure 6: This figure seems superfluous; it only gives the number of nodes = 569 concepts as (could be better if needed) described in the text. (the “leaf nodes” and “branches” are not mentioned in the text, but could be without the figure if needed)
Indeed, there is no real added value in this figure. The figure has been removed.
> l.357: “…as a separate ontology module,..”
An article has been added.
> Chap 3.8.1. It may be that the first paragraph in this section belongs to the previous section and the last paragraph to the next section, before Fig. 10 ?
The idea was that the first paragraph provides the terminology used in the second paragraph; and, in fact, the section title was misleading. The section has now been rephrased and restructured, hopefully making it clearer. Also, the section title has been changed.
> l.430: Were “few of the violations eliminated”, or “were there few violations and they have been eliminated” ?
The latter is true. The sentence has been rephrased for clarification.
> l.446, 463: Figure notations (Fig 13) and (Fig 14) are in different subsections than the figures which is unfortunate for the reading. This is also the case with other figures. If possible, figures should be inserted where noted in txt. (Some journals change figure placement to first figure notation in the text in the final editing..)
The LaTeX style seems to give overall layout higher precedence over optimal location of the figures with respect to their location of reference. The revised version now contains various tweaks to enforce better placement of the figure, though a minimal change of the text in the course of further editing may again break the placement. Probably, placement of figures should be checked once again as last step of editing.
> Figure 17: The fig. should probably not be inside the topics list (?)
The previous comment also applies here: With help of various tweaks, Fig. 17 is now no more inside of a topic list (as well as some more figures), though even a small editorial change may well break things again. Probably, placement of figures should be checked once again as last step of editing.
Reviewer 3 Report
The article presents an interesting and useful issue, which is how to develop an ontology for the protection of cultural heritage. However, some improvements are suggested.
The paper should include the contributions of the panel of experts in order to better understand and identify their role.
It is recommended in section 3.4 to provide another case as an example in order to better understand the variability of cases that can be found.
The methodology should say how many assets the KERES project has been built on.
There are some stylistic issues that should be improved, especially to fit a scientific language. Currently, when reading the article, it seems that we are dealing with a text for commercial purposes and not a scientific text. A general review is recommended in this regard. There is an excessive abundance of first person pronouns like “we” or “our”, it is not necessary. It is evident that it is yours because you are the authors. Revise the text in this regard by removing these unnecessary pronouns. An example, in Figure 3 it says “Our Methodology for Designing and Implementing the KERES Ontology” and could say “Methodology for Designing and Implementing the KERES Ontology”.
In a scientific text, colloquial expressions such as “the tree cadaster blows up the ontology” should not be used. Or make statements like “none of them actually can compete with the broad range of areas that we address in the KERES project”
Neither should there be references in the abstract, these should only appear in the body of the text.
The paragraphs should not be interrupted by the images and revise the font of the text, which is always the same.
Changes are recommended to the following images:
- Figure 1. It is recommended to move it later in the text, to when the ontology has already been explained and use it there as an example. As it is now in the text, it is not understandable because, moreover, the information it shows is partial.
- Figure 4. Improve quality, a title appears that is too big, the German part is either translated or removed. Improve the image that is pixelated
- Figure 7. Translate into English the part that remains in zoom
- Figure 22. A Lorem Ipsum appears, review it.
Finally some language issues:
- Page 2, first sentence, rewrite it, it's too long.
- Page 4 “there exists a large number of individual efforts” should say “there exist a large number of individual efforts!”
- Page 4 “In contrast, the KERES project builds on an elaborate hierarchy of ontological concepts that cover a wide range of cases with a plethora…” should say “covers”
- Line 429, there is a duplicate parenthesis
Author Response
> The paper should include the contributions of the panel of experts in order to better understand and identify their role.
Done: The experts' contributions have been added to Sect. 3.5.
> It is recommended in section 3.4 to provide another case as an example in order to better understand the variability of cases that can be found.
Done: The Franconian Open-Air Museum has been added as another example.
> The methodology should say how many assets the KERES project has been built on.
Done: Sect. 3.3, Purpose and Scope, has been extended accordingly.
> There are some stylistic issues that should be improved, especially to fit a scientific language. Currently, when reading the article, it seems that we are dealing with a text for commercial purposes and not a scientific text. A general review is recommended in this regard. There is an excessive abundance of first person pronouns like “we” or “our”, it is not necessary. It is evident that it is yours because you are the authors. Revise the text in this regard by removing these unnecessary pronouns. An example, in Figure 3 it says “Our Methodology for Designing and Implementing the KERES Ontology” and could say “Methodology for Designing and Implementing the KERES Ontology”.
Done: The overall number of person pronouns has been substantially reduced.
> In a scientific text, colloquial expressions such as “the tree cadaster blows up the ontology” should not be used. Or make statements like “none of them actually can compete with the broad range of areas that we address in the KERES project”
Done: The affected text has been rephrased to avoid unappropriate language.
> Neither should there be references in the abstract, these should only appear in the body of the text.
Done: The reference has been removed from the abstract.
> The paragraphs should not be interrupted by the images and revise the font of the text, which is always the same.
Done: The images are now differently placed. Also, various font selection errors have been fixed.
> Changes are recommended to the following images:
>
> - Figure 1. It is recommended to move it later in the text, to when the ontology has already been explained and use it there as an example. As it is now in the text, it is not understandable because, moreover, the information it shows is partial.
Done: Fig. 1 has been removed from the introduction.
> - Figure 4. Improve quality, a title appears that is too big, the German part is either translated or removed. Improve the image that is pixelated
Done: The screenshot now shows the English version of the web page and with higher screen resolution, and the caption mentions that it is a screenshot.
> - Figure 7. Translate into English the part that remains in zoom
Done: Translation has been added.
> - Figure 22. A Lorem Ipsum appears, review it.
Done.
> Finally some language issues:
>
> - Page 2, first sentence, rewrite it, it's too long.
Done.
> - Page 4 “there exists a large number of individual efforts” should say “there exist a large number of individual efforts!”
Done: Checked against lexeme 1e of lemma “number” in Webster's Third New International Dictionary.
> - Page 4 “In contrast, the KERES project builds on an elaborate hierarchy of ontological concepts that cover a wide range of cases with a plethora…” should say “covers”
Done: Checked: The ontological concepts cover a wide range of cases.
> - Line 429, there is a duplicate parenthesis
Done: The duplicate parenthesis has been eliminated.
Round 2
Reviewer 3 Report
The suggestions and concerns of the original review have been addressed within the revised paper.